# Awe and Prosocial Behavior: The Mediating Role of Presence of Meaning in Life and the Moderating Role of Perceived Social Support

**DOI:** 10.3390/ijerph19116466

**Published:** 2022-05-26

**Authors:** Ya-Nan Fu, Ruodan Feng, Qun Liu, Yumei He, Ofir Turel, Shuyue Zhang, Qinghua He

**Affiliations:** 1Department of Psychology, Faculty of Education, Guangxi Normal University, Guilin 541004, China; fuyanan1210@163.com (Y.-N.F.); frd1014@163.com (R.F.); njtcliuqun@outlook.com (Q.L.); tastyandtree@163.com (Y.H.); 2Guangxi University and College Key Laboratory of Cognitive Neuroscience and Applied Psychology, Guangxi Normal University, Guilin 541004, China; 3School of Marxism, Neijiang Normal University, Neijiang 641112, China; 4School of Computing and Information Systems, The University of Melbourne, Parkville, VIC 3010, Australia; oturel@fullerton.edu; 5MOE Key Laboratory of Cognition and Personality, Faculty of Psychology, Southwest University, Chongqing 400715, China

**Keywords:** awe, prosocial behavior, presence of meaning in life, perceived social support, longitudinal moderated mediation model

## Abstract

Although awe has been shown to increase prosocial behavior, there is limited knowledge about the mechanisms underlying this relationship, and about this relationship during unique periods. To bridge these gaps, this study examined the influence of awe on prosocial behaviors, the mediating role of the presence of meaning in life, and the moderating role of perceived social support. Based on longitudinal surveys from 676 Chinese college students we showed that: (1) awe was positively associated with prosocial behavior; (2) the presence of meaning in life mediated this association, and; (3) these associations were moderated by perceived social support. Specifically, the positive relationship between awe and the presence of meaning in life was only significant for college students with low perceived social support; and the positive relationship between the presence of meaning in life and prosocial behavior was stronger for college students with high perceived social support.

## 1. Introduction

Major disaster events and ongoing pandemics can have important social, psychological, and health effects on individuals. Such outcomes in the face of COVID-19, including in college students, are yet to be fully understood. Prosocial behaviors include overt actions intended to benefit others [1]. They are important because prosocial behaviors such as cooperating, sharing, donating, and helping others contribute to social harmony [2] and afford the human race to thrive [3]. Here, we focus on factors that might have affected students’ prosocial behavior, because such behaviors have major societal and individual implications [4,5].

Given its importance, researchers have studied various correlates of prosocial behavior, including attributes of the helper and the recipient, situational factors, and sociocultural factors. For example, a study of children and adolescents found that prosocial behavior is associated with positive emotions [6]. Such positive emotional states remind participants of positive experiences and encourage similar actions in the future, such as prosocial behavior [7]. This is in line with the social outlook model, which suggests that positive emotions lead to more positive social cognition, which in turn leads to greater attention to prosocial behavior [8].

Awe is an important positive emotion disposition [9]. It is often conceptualized as the emotional experience of an individual in the face of a vast, powerful stimulus that goes beyond the existing cognitive framework [10]. It is also an important driver of prosocial behavior: the induction of awe, compared to the induction of amusement or a neutral condition, leads to increased prosocial behavioral intentions of generosity and helping a person in need [11].

Although people with a high level of awe are more likely than others to prefer and choose prosocial behavior, much less is known about how and when awe increases prosocial behavior. Thus, the present study first aims to validate the relationship between awe and prosocial behavior (i.e., replicate prior research findings). Second, it aims to extend previous literature by theorizing and testing the roles of the presence of meaning in life and perceived social support in translating awe into prosocial behavior. To this end, the study utilizes a sample of college students and examines the mediating effect of the presence of meaning in life and the moderating effect of perceived social support.

### 1.1. Awe and Prosocial Behavior

Awe is a complex emotion that encapsulates many feelings such as confusion, admiration, surprise, and obedience. Its importance stems from the fact that it has a range of positive effects on individuals, leading to less self-concern and more prosocial behavior [11,12]. The induction of awe has been supported to lead to more prosocial behavior than other more frequently studied varieties of positive emotions. With awe, individuals will feel small, focus on something greater than the self, forget daily concerns, have a close connection with the surrounding world, and hope to prolong or memorize the experience [9]. The subjective experience of awe is consistent with the notion of self-transcendence [13], and it is fundamentally out of concern to enhance the welfare of others, thus promoting prosocial behavior [14]. Studies have reported that awe can significantly and positively influence prosocial behavior tendency and online altruism [15]. Therefore, this study will verify this finding.

### 1.2. Presence of Meaning in Life as a Mediator

The presence of meaning in life refers to the extent to which people comprehend, make sense of, or see the significance in their lives, accompanied by the degree to which they perceive themselves to have a purpose, mission, or overarching aim in life [16]. We postulate that the presence of meaning in life can mediate the expected association between awe and prosocial behavior.

The reason is that firstly, awe can enhance an individual’s presence of meaning in life. Indeed, an individual’s positive emotion has a significant positive correlation with meaning in life [17], and especially with the presence of meaning in life [18]. This happens because individuals with higher positive emotions are usually more likely to experience a higher level of meaning in life. Awe, as a kind of self-transcendent positive emotion, has also been supported to promote meaning in life. Using video to initiate awe, the study found that awe can induce a positive feeling and ultimately increase meaning in life [19].

Second, the presence of meaning in life can positively influence prosocial behavior [20]. The higher level of meaning in life, the better life satisfaction, less psychological distress and negative emotions, and more prosocial behavior [21]. This association is rooted in self-determination theory, which postulates that people with a higher meaning in life, driven by their own goals, missions, and tasks, have stronger intrinsic motivations for prosocial behaviors, and will engage in prosocial behaviors autonomously and voluntarily [22].

Thus, it is possible that awe, which is accompanied by a strong emotional experience of self-transcendence that enables individuals to pursue spirituality, can increase the sense of meaning in life in individuals [11,23,24]. Furthermore, a strong meaning in life can encourage individuals to pay more attention to the current difficulties and needs of others [25] and through this, also promote social activities that transcend the meaning in life [26], such as spiritual satisfaction by helping others or engaging in charity. Indeed, dispositional awe, which is a personality trait that is stable across time and situation, can not only directly influence prosocial behavior but can indirectly influence prosocial behavior by enhancing self-transcendence meaning in life [27]. Lin reveals the mediating role of self-transcendent meaning in life between dispositional awe and prosocial behavior in his article but argues that longitudinal study may be more convincing [28]. Based on such findings, the mediation mechanism we put forth can shed light on the important association between awe and prosocial behavior. The abovementioned literature has led us to hypothesize that:

**Hypothesis** **1** **(H1).**
*The presence of meaning in life mediates the effect of awe on prosocial behavior.*


### 1.3. Perceived Social Support as a Moderator

Awe can be significantly related to prosocial behavior through the mediator of the presence of meaning in life, but not all individuals being in awe will equally follow this process and engage in more prosocial behavior. Therefore, it is important to explore those factors that may moderate the strength of the relationships among awe, the presence of meaning in life, and prosocial behavior. We hypothesize that perceived social support can be such a factor through its protecting/sheltering effect. The definition of perceived social support is the existence of support resources when they are needed [29]. Individuals can gain perceived social support from families, peers, and other people who may provide the needed support when a person seeks it.

We argue that perceived social support can moderate the effect of awe on the presence of meaning in life. Study shows that the more social support individuals get from family and friends, the higher meaning in life they will have [30]. Emotion has been reported to associate with social support [31]. Greater social support was associated with lower levels of loneliness overall, but the receipt of social support did not modify one’s expression of loneliness over time [32]. Awe, as a kind of positive emotion [9], might also be associated with perceived social support. Importantly, awe has complicated effects on meaning in life [19]. Many awe experiences may contain a positive flavor, which contributes to awe and generally positive emotions, thus having a positive impact on meaning in life [19]. At the same time, however, awe experiences lead to a diminished self that reflects feelings of smallness and insignificance, which might negatively influence meaning [19]. These complicated effects may suggest a potential moderating mechanism [19]. Perceived social support is a protective factor that reduces the sensitivity of meaning in life to negative predictors and increases the effect of positive predictors [19,30,33]. Therefore, we suggest that perceived social support may moderate the association between awe and the presence of meaning in life.

The direction of moderation, though, is not clear, as we explain below. On the one hand, the protective-enhancing hypothesis maintains that different protective factors interact to affect the development of individuals and that one protective factor (i.e., perceived social support) will enhance the function of the other protective factor (i.e., awe) [34,35]. Due to this hypothesis, rather than low levels of perceived social support, the promoting effect of awe on the presence of meaning in life would be stronger for people with high perceived social support, implying that interventions aimed at improving awe would be particularly beneficial for people who score high in perceived social support. On the other hand, the protective-attenuating hypothesis holds that one protective factor usually takes advantage and this beneficial impact is especially potent when individuals possess low levels of another protective factor [34,36]. According to this perspective, the promoting effect of awe on the presence of meaning in life would be stronger for people with low rather than high levels of perceived social support, suggesting that interventions seeking to increase awe would be more effective for people who possess low perceived social support. Considering that the absence of empirical studies makes it impossible for us to infer clearly which hypothesis will hold, in the present study we suggest competing hypotheses. To see which one holds, we will explore the moderating pattern of perceived social support between awe and the presence of meaning in life.

Furthermore, perceived social support serves as an important environmental factor that may moderate the effect of the presence of meaning on prosocial behavior. With the rise of ecological systems theory, researchers have realized that individual development is the result of the interaction between individual characteristics and environmental factors. The individual-environment interaction model suggests that an individual’s behavior is formed and developed through the interaction between the individual and the environment, and individual factors interact with environmental factors to affect individual development [37]. Consequently, perceived social support, as an environmental factor [38] may interact with the presence of meaning in life, thus affecting the individual behavior choices. Such mechanisms are not yet fully understood. Only one recent study indicates that peer support moderates the relationship between meaning in life and prosocial behavior [39], the effect of meaning in life on prosocial behavior is more significant among college students with a low level of peer support than those with a high level of peer support. Drawing upon such evidence, we propose the second hypothesis:

**Hypothesis** **2** **(H2).**
*Perceived social support moderates the relationship between awe and the presence of meaning in life and between the presence of meaning in life and prosocial behavior.*


### 1.4. The Present Study

The integration of the hypotheses suggests a moderated mediation model linking awe to prosocial behavior, in which the presence of meaning is a mediator and perceived social support is a moderator (Figure 1). We focus on these associations during the COVID-19 pandemic, given the ability of major events to influence people’s psychological and behavioral processes. We use a longitudinal design to test the hypotheses with samples of students as students have been shown to be adversely affected by the pandemic [40,41,42,43], the social isolation it has brought, and the way it altered the presence of meaning in life, perceived social support, and possibly awe in people.

## 2. Materials and Methods

### 2.1. Participants and Procedure

Using the method of cluster sampling, students from 22 classes in a university in China were invited to participate. Participants completed a survey containing all relevant scales. All procedures were reviewed and approved by the local ethics committee. The study was conducted in February (T1) and July (T2) 2020. In February, 686 students from the 702 total consenting participants completed the Awe subscale of the Dispositional Positive Emotion Scales. After six months, 676 students from the 686 students completed the Prosocial Tendencies Measure, the Presence of Meaning in Life Subscale, and the Perceived Social Support Scale (98.54% retention rate) at Time 2. Finally, a total of 676 students (508 females) participated in the current study (*M*_age_ = 20.66 ± 1.09 years old). The effective response rate was 96.30%.

### 2.2. Materials

#### 2.2.1. Awe

The awe subscale included 6 items which were rated on a 7-point Likert scale. The higher the score, the higher the degree of awe. It was a subscale of the Dispositional Positive Emotion Scales (DPES) questionnaire [44]. Cronbach α for this subscale was 0.78 in the current study.

#### 2.2.2. Prosocial Behavior

The Prosocial Tendencies Measure (PTM) was compiled by Carlo et al. [3] and revised by Kou et al. [45]. The scale consisted of 26 items, which measured prosocial behavior in six situations: public, anonymous, altruistic, compliant, emotional, and urgent. It used a 5-point Likert scale. The higher the total score, the more prosocial behavior people presented. For the current study, the Cronbach α was 0.96.

#### 2.2.3. The Presence of Meaning in Life

The Presence of Meaning in Life Subscale of the Chinese version of the meaning in life questionnaire was compiled by Steger [16] and revised by Zhang and Xu [46]. There were 5 items in the subscale, rated on a 7-point Likert scale. The higher the score, the higher the degree of meaning in life. In this study, the Cronbach α for this subscale was 0.77.

#### 2.2.4. Perceived Social Support

The Chinese version of the Perceived Social Support Scale (PSSS) was compiled by Blumenthal et al. [47] and revised by Huang et al. [48]. The scale had 12 items to measure the degree of support from family, friends, and others. The total score reflected the total degree of social support. Items were rated on a 7-point Likert scale. Cronbach α for this scale was 0.96.

### 2.3. Data Analysis

We used SPSS 25 for data preprocessing, description statistics, and basic analyses. We also used the SPSS PROCESS macro by Hayes [49] for testing the moderated mediation model. Model 4 of the PROCESS macro was used to test the mediating effect of the presence of meaning in life in the relationship between awe and prosocial behavior. Model 58 of the PROCESS macro was employed to test the moderated mediating effect.

## 3. Results

### 3.1. Common Method Deviation Test

We used Harman’s single-factor test to check for common method variance bias [50]. The result showed that there were eight factors with eigenvalues greater than 1, with the largest single factor explaining 36.25% of the variance, which was less than the judgment standards of 40% [50]. Therefore, common method bias was not considered to be a major threat in this study.

### 3.2. Descriptive Statistic and Correlations

Correlation analysis was performed with the mean scores of the awe, prosocial behavior, perceived social support, and the presence of meaning in life scales in both T1 and T2. Pearson’s correlations and descriptive statistics for the model’s variables are reported in Table 1. The results showed that all variables were significantly correlated.

### 3.3. Testing for Mediation Effect

Model 4 of the PROCESS macro [49] was adopted to examine the mediating effect of the presence of meaning in life between awe and prosocial behavior after controlling gender. As shown in Model 2 of Table 2, awe T1 positively predicted the presence of meaning in life T2 (*β* = 0.19, *t* = 5.09, *SE* = 0.04, *p* < 0.001). In Model 3, when controlling gender and the presence of meaning T2, it positively predicted prosocial behavior T2 (*β* = 0.40, *t* = 11.25, *SE* = 0.04, *p* < 0.001), whereas the direct effect of awe on prosocial behavior was substantially reduced (*β* = 0.11, *t* = 3.06, *SE* = 0.04, *p* = 0.002). The indirect effect of awe on prosocial behavior via the presence of meaning in life was significant (*β_indirect_* = 0.08, Boot *SE* = 0.02, *p* < 0.001, 95% Boot *CI* [0.05, 0.12]), supporting the mediation claims. The mediation effect accounted for 56.44% of the total effect of awe on prosocial behavior. Therefore, Hypothesis 1 was supported.

### 3.4. Moderated Mediation Effect Analysis

Model 58 of the PROCESS macro [49] was used to test whether perceived social support moderated the relation between awe and the presence of meaning in life (first-stage moderation), as well as the link between the presence of meaning in life and prosocial behavior (second-stage moderation) after controlling gender.

In the first-stage moderation, as shown in Model 1 in Table 3, awe T1 was positively associated with the presence of meaning in life T2 (*β* = 0.09, *t* = 2.92, *SE* = 0.03, *p* = 0.003), and this association was moderated by perceived social support T2 (*β* = −0.08, *t* = −2.88, *SE* = 0.03, *p* = 0.004, 95% *CI* [−0.14, −0.03]). The simple slope test in Figure 2 indicated that for college students with low perceived social support (1 *SD* below the mean), awe significantly predicted prosocial behavior (*β_simple_* = 0.18, *SE* = 0.04, *p* < 0.001, 95% *CI* [0.09, 0.27]). However, for college students with high perceived social support (1 *SD* above the mean), the relationship between awe and the presence of meaning in life became non-significant (*β_simple_* = 0.01, *SE* = 0.04, *p* = 0.82, 95% *CI* [−0.07, 0.09]).

In the second-stage moderation, as presented in Model 2 in Table 3, the presence of meaning in life T2 was positively associated with prosocial behavior T2 (*β* = 0.19, *t* = 4.92, *SE* = 0.04, *p* < 0.001), and this association was moderated by perceived social support T2 (*β* = 0.06, *t* = 2.17, *SE* = 0.03, *p* = 0.03, 95% *CI* [0.01, 0.11]). The simple slope test in Figure 3 showed that the association between the presence of meaning in life and prosocial behavior was significantly weaker for participants with low levels of perceived social support (1 *SD* below the mean) (*β_simple_* = 0.14, *SE* = 0.04, *p* = 0.003, 95% *CI* [0.05, 0.22]), whereas this positive association was much stronger for participants with high levels of perceived social support (1 *SD* above the mean) (*β_simple_* = 0.25, *SE* = 0.05, *p* < 0.001, 95% *CI* [0.15, 0.35]).

The bias-corrected percentile bootstrap analyses further showed that the indirect effect of awe on prosocial behavior through the presence of meaning in life was moderated by perceived social support. As shown in Table 4, the indirect association between awe and prosocial behavior is significantly stronger for college students with a low level of perceived social support (*β* = 0.01, *SE* = 0.01, 95% *CI* [−0.02, 0.04]), whereas this indirect relationship was not significant for college students with high levels of perceived social support (*β* = 0.003, *SE* = 0.011, 95% *CI* [−0.020, 0.024]). These results supported Hypothesis 2. In sum, it indicated that perceived social support moderated the indirect associations between awe and prosocial behavior via the presence of meaning in life.

## 4. Discussion

In this study, we explored the mechanisms that translate awe into prosocial behavior. Although the effect of awe on prosocial behavior has received considerable empirical support [12,14,27], the potential mediation and moderation mechanisms are unclear. Thus, we put forth a moderated mediation model to test how awe works and whether all individuals are equally influenced by awe. Our study showed that awe was significantly and positively associated with prosocial behavior. We further revealed that the presence of meaning in life partially mediated this relationship. Furthermore, perceived social support moderated the relationship between awe and the presence of meaning in life as well as the presence of meaning in life and prosocial behavior.

### 4.1. Awe and Prosocial Behavior

The results of this study show that the higher the level of awe is, the higher the prosocial behavior is, which is consistent with the results of previous studies [12]. These results indicate that awe is an important factor affecting prosocial behavior. Awe is a type of self-transcendent positive emotion disposition [9,14]. Self-transcendence usually comes from other focused assessments, shifting attention to the needs and concerns of others rather than to themselves. As such, self-transcendence emotions are others-oriented, which reduces people’s attention to themselves and drives increased sensitivity to others. These attributes motivate prosocial behavior. This is also consistent with the theory of values [51]. Values guide people’s behavior and are the basic beliefs of individuals before events. Self-transcendental values emphasize reducing self-importance and increasing attention to others and nature; values, attitudes, and behaviors have the same motivational basis [52]. Therefore, self-transcendental emotions, including awe can lead to reduced self-attention and increased prosocial behavior.

### 4.2. The Mediating Role of Presence of Meaning in Life

As predicted, the presence of meaning in life partially accounted for the association between awe and prosocial behavior among college students. Therefore, the presence of meaning is not only an outcome of awe but also a catalyst of prosocial behavior. As far as we know, this is the first study that uses a longitudinal design to examine the mediating role of the presence of meaning in life in the relationship between awe and prosocial behavior among college students.

In addition to the overall mediation result, each separate link in our mediation model deserves attention. For the first stage of the mediation model (i.e., awe→presence of meaning in life), we found a positive association. This finding coincides with the previous studies [19]. The presumed increase of an individual’s sense of awe is accompanied by a strong experience of self-transcendence, which enables individuals to pursue the spiritual world and have a strong presence of meaning in life. For the second stage of the mediation process (i.e., presence of meaning in life→prosocial behavior), the present study found a positive association. This finding supports the self-determination theory logic [22]. Under the threat of the COVID-19 pandemic, new goals (e.g., cherishing life, serving society) have emerged in people. Driven by such goals, college students with a high level of the presence of meaning presented a stronger intrinsic prosocial tendency. Because the presence of meaning in life drives individuals to focus on the needs of others [25], they were, as supported by our results, more likely to engage in prosocial behavior.

This study not only explains how awe affects prosocial behavior from a new theoretical perspective but also expands the research field of prosocial behavior by accounting for the above-discussed mediation mechanism. Our findings imply that in order to increase prosocial behavior, we must improve individual notions, induce positive emotions and specifically awe, and improve the presence of meaning in life.

### 4.3. The Moderating Role of Perceived Social Support

The results also showed that perceived social support moderated the relationship between awe and the presence of meaning as well as the presence of meaning and prosocial behavior. The relationship between awe and the presence of meaning in life was only significant for college students with low perceived social support. The relationship between the presence of meaning in life and prosocial behavior was also moderated by perceived social support, and this relationship was stronger for college students with high rather than low perceived social support. These results show that the influence of awe may not be the same at all levels of perceived social support. The promoting effect of awe on prosocial behavior is contingent on one’s level of perceived social support.

For college students with low perceived social support, awe plays a greater role in promoting the presence of meaning in life and thus increasing their prosocial behavior. This result suggests that improving the awe of college students with low perceived social support can effectively increase their prosocial behavior. However, for college students with high perceived social support, the beneficial effect of awe is weakened. According to the “shock absorber” model of social support, perceived social support can enhance positive emotions and reduce other negative effects [53]. Therefore, one explanation is that appropriate and moderate perceived social support can improve an individual’s happiness and life satisfaction. College students with high levels of perceived social support are more likely to be in awe, and the presence of meaning can also be improved accordingly. However, when perceived social support is too high, the presence of meaning caused by awe may reach a bottleneck. As a result, no matter how the level of awe changes, the presence of meaning will remain high due to the high level of perceived social support and will be only marginally affected by the change of awe. The findings of this study further show that college students with high levels of perceived social support, are still more likely to be involved in prosocial behavior. If they have moderate awe working in conjunction with high levels of perceived social support, it would be more effective to maximize their prosocial behavior. Note that perceived social support “attenuates” the relationship between awe and the presence of meaning, thus providing support to the protective-attenuating hypothesis [34,36]. This result still implies that perceived social support is a protective factor for college students’ prosocial behavior because college students with high perceived social support exhibit significantly higher levels of the presence of meaning than those with low perceived social support. In addition, this also reminds us that for sensitive groups with a low level of social support, we should induce their level of awe as much as possible to increase their presence of meaning in life.

Furthermore, in line with the individual-environment interaction model [37], as an environmental factor, perceived social support increases the positive effect of the presence of meaning in life on prosocial behavior. The effect of the presence of meaning in life on prosocial behavior is stronger for college students with high levels of perceived social support. One possible account is that for college students with a high level of perceived social support, when they lack meaning, they can get comfort, encouragement, and company from families and friends to compensate for the lack of presence of meaning in life. In contrast, college students with a low level of perceived social support, may feel excluded from social relationships with others and reduce their caring about other things, such as learning burnout [54], academic burnout [55,56], and sport burnout [57].

### 4.4. Limitations and Future Directions

The results of this study have some implications for the promotion of prosocial behavior: awe is not only about nature, but also about life and global issues, effort, and how humanity gets together to resolve complex issues. When we are in awe, it will also increase our presence of meaning in life, which will significantly predict prosocial behavior. Therefore, we can increase people’s prosocial behavior by increasing their awe and presence of meaning in life to make the society we live in more harmonious. It is also a valuable perspective to bring in more variables around mental health and to explore the potential mechanisms between awe and prosocial behavior. At the same time, this study also has some limitations. Specifically, we use self-reports from one country and one age segment. Future research can collect data from multiple information sources such as parents, teachers, and peers and extend the generalizability of our findings by collecting data in other countries and in different age groups. Secondly, this study was assessed at two time points, and a three-wave longitudinal design may be useful to confirm our findings. Thirdly, when we selected the subjects by cluster sampling, we did not predict in advance that the number of male and female participants in these schools was so unbalanced, so the external validity of the results may be insufficient. Future research can reaffirm our model in more gender-balanced samples.

## 5. Conclusions

In summary, this study is an important step in unpacking how awe relates to follow-up prosocial behavior. This study found that: (1) awe can significantly positively predict prosocial behavior and can also indirectly predict prosocial behavior through the presence of meaning in life; (2) these associations are moderated by the perceived social support. Specifically, the positive relationship between awe and the presence of meaning in life was only significant for college students with low perceived social support; and the positive relationship between the presence of meaning in life and prosocial behavior was stronger for college students with high perceived social support.

## Figures and Tables

**Figure 1 ijerph-19-06466-f001:**
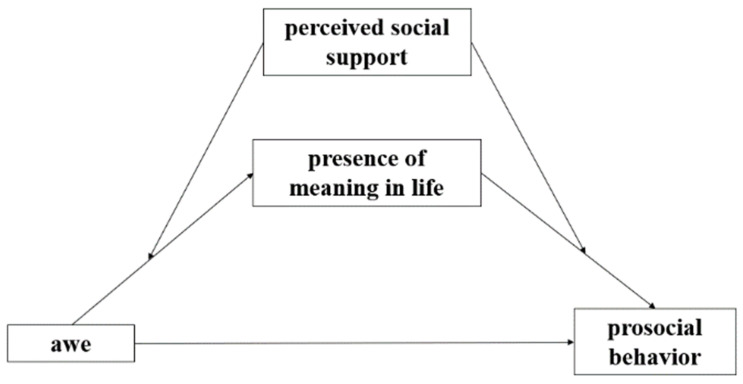
The proposed theoretical model.

**Figure 2 ijerph-19-06466-f002:**
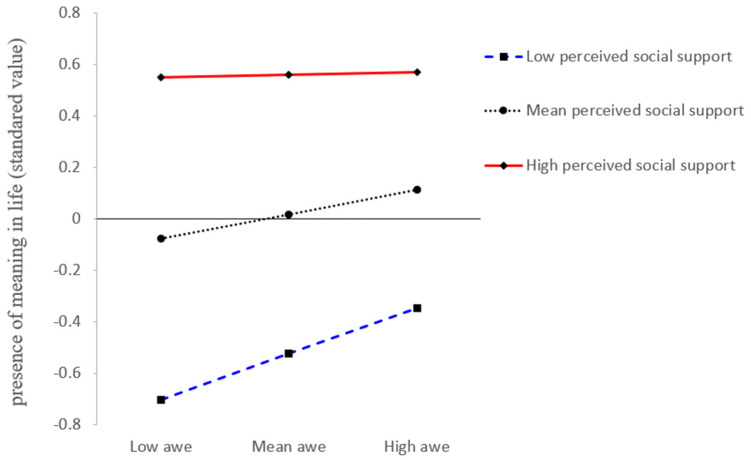
Interaction Plot: Perceived social support moderates the effect of awe on the presence of meaning in life.

**Figure 3 ijerph-19-06466-f003:**
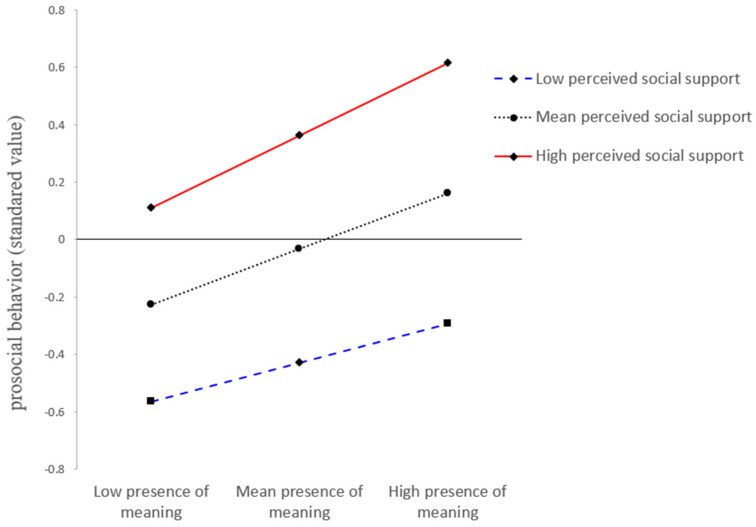
Interaction Plot: Perceived social support moderates the effect of the presence of meaning in life on prosocial behavior.

**Table 1 ijerph-19-06466-t001:** Means, standard deviations (SD), and correlations of study variables (*N* = 676).

Variable	A T1	PB T2	PS T2	MP T2
A T1	1.00			
PB T2	0.192 ***	1.00		
PS T2	0.218 ***	0.519 ***	1.00	
MP T2	0.209 ***	0.423 ***	0.560 ***	1.00
*M*	29.61	93.89	58.60	24.72
*SD*	5.75	16.21	13.35	5.14

Note. *M* is Mean, *SD* is standard deviation. *** *p* < 0.001. A = awe, PB = prosocial behavior, PS = perceived social support, MP = presence of meaning in life.

**Table 2 ijerph-19-06466-t002:** Testing the mediation effect of the presence of meaning in life.

Predictors	Model 1 (PB T2)		Model 2 (MP T2)		Model 3 (PB T2)	
	*β*	*t*	*β*	*t*	*β*	*t*
gender	0.04	0.48	0.03	0.92	0.004	0.13
A T1	0.19	5.09 ***	0.21	5.57 ***	0.11	3.06 **
MP T2					0.40	11.25 ***
*R^2^*	0.04		0.05		0.19	
*F*	13.04 ***		15.86 ***		52.48 ***	

Note. *N* = 676. ** *p* < 0.01. *** *p* < 0.001. A = awe, PB = prosocial behavior, MP = presence of meaning in life.

**Table 3 ijerph-19-06466-t003:** Testing the moderated mediation effect of perceived social support.

Predictors	Model 1 (MP T2)		Model 2 (PB T2)	
	*β*	*t*	*β*	*t*
gender	0.03	0.37	0.02	0.27
A T1	0.09	2.92 **	0.07	2.09 *
PS T2	0.54	16.77 ***	0.40	10.07 ***
A T1 × PS T2	−0.08	−2.88 **		
MP T2			0.19	4.92 ***
MP T2 × PS T2			0.06	2.17 *
*R* ^2^	0.33		0.30	
*F*	82.74 ***		58.46 ***	

Note. *N* = 676. * *p* < 0.05. ** *p* < 0.01. *** *p* < 0.001. A = awe, PB = prosocial behavior, PS = perceived social support, MP = presence of meaning in life.

**Table 4 ijerph-19-06466-t004:** Conditional indirect effect of awe on prosocial behavior through the presence of meaning in life at different levels of perceived social support.

Level of PSS	Indirect Effect	Boot *SE*	Boot LLCI	Boot ULCI
Low (Mean − 1 *SD*)	0.024	0.012	0.006	0.050
Mean	0.018	0.008	0.004	0.036
High (Mean + 1 *SD*)	0.003	0.011	−0.020	0.024

Note. *N* = 676. *SD* is standard deviation. PSS = perceived social support.

## Data Availability

The datasets generated during the current study that support the findings of this study are available from the corresponding author upon reasonable request.

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
