# Peer review of "Awe and Prosocial Behavior: The Mediating Role of Presence of Meaning in Life and the Moderating Role of Perceived Social Support"

_ijerph, 2022, doi:10.3390/ijerph19116466_

Round 1
Reviewer 1 Report
Dear authors,
Thank you for taking the comments on board - the piece is clearer and has been strengthened. A couple of minor suggestions:
93 it is not clear what is meant by 'the mental world' - I think you could say 'spirituality' instead
100 it is unclear what is meant by 'future time perspective' which means that for the reader, this paragraph becomes incoherent. please define this term for clarity
123 typo - change 'have' to 'has'
173-175 as you are making a claim in this sentence I would expect to see one or more citations backing this up, otherwise it is unconvincing
Please see my previous feedback with regard to the set of questionnaires used:
2.2 Materials - can you say something about the robustness of the suite of tests you administered? why this specific group of tests? how do we know that together this is a reliable and sound mix? is there consensus in the field that these are the preferred tests to use?
297 add an 's' to mechanism
407 add an 's' to sample
Reviewer 2 Report
The authors addressed my previous concerns. I consider the manuscript suitable for publication.
Author Response
Please see the attachment.

This manuscript is a resubmission of an earlier submission. The following is a list of the peer review reports and author responses from that submission.
Round 1
Reviewer 1 Report
Dear authors,
I think your work is very interesting and could set the foundation for further work that supports people to act in more empathic and helpful ways toward others. However, the article needs to be revised, tightened and strengthened in parts.
Wishing you all the best and I hope the below revisions help you to further develop your piece.
29-35 Suggest rewriting the first paragraph of the introduction as it is a bit clumsy
47 it is not clear what is meant by 'the existing cognitive framework'
54 'replicate prior insights' - does this mean replicate prior research findings?
64 'more than other positive emotions' - this is too vague and needs clarifying
65 describing the world as 'much more important and admirable' is an unusual turn of phrase that does not add much. Suggest rephrasing
66 'organized; does not seem to be the right word here - I don't think people organize their awe. Suggest rephrasing
68 change 'supported' to 'found' or 'reported'
78-79 rephrase this sentence as it is unclear
81 how is it special? what do you mean here?
82 this sentence needs rephrasing as it is unclear - this is also the first time that video is mentioned and there is no context for it
85 the first 'higher' needs to be replaced with another word as it does not make sense
92 what is meant by the spiritual world? do you mean spirituality? the spiritual world usuall refers to good and evil spirits - i think you may mean spirituality?
95 it is not clear what is meant by 'social activities that transcend the meaning in life'
Some of the content in the above sections could be considered quite abstract concepts so it will help the reader immensely if there is greater coherence, accurate terms used, and definitions and or / context provided where necessary.
96 define the term 'dispositional awe'
97 what is meant by 'an individual's future time perspective'? it is unclear
100 which longitudinal study are you referring to?
101 typo - fort should be forth
100-104 this paragraph reads as a collection of disparate ideas and is not coherent - please revise it
1.3 starts very well
116 their - who? it is not clear
123 change 'supported' to 'found' or 'reported'
127 change 'think' to 'suggest' or 'hypothesise'
171 a short-term longitudinal design seems tautological - can you please clarify whether it is short term or longitudinal?
173 I imagine there are other reasons to have chosen students - possibly access reasons. I think it would be good to explain this, as surely there are many other groups in society that are experiencing the exact same conditions as a result of COVID and could also benefit from studies that address helping behaviour and so on?
187 over 75% of your sample identified as female. as a reader i am wondering whether the previous studies that you cite also have a bias toward women / girls?
This may also be listed as a limitation of the current study as it is not likely to be completely representative and future work could build on your results, addressing this
2.2 Materials - can you say something about the robustness of the suite of tests you administered? why this specific group of tests? how do we know that together this is a reliable and sound mix? is there consensus in the field that these are the preferred tests to use?
304-310 you are making a lot of unsubstantiated claims here - i think you need to back these up with citations from the literature or other sources if peer-reviewed literature is not yet available.
I am unsure what you mean about the tug-of-war between medical staff and the virus or how this is connected directly with awareness of social ties - suggest rephrasing for clarity. What is meant by awareness of social ties? Perhaps it is not awareness but the actual experience of isolation, quarantine, ill health (etc)
315 i don't think you can generalise your results to all Chinese college students as the sample is so small and not representative - maybe better to conclude that among your sample, awe was....
323-324 repetitive with earlier text
345-348 I am not convinced that the pandemic has increased people's sense of meaning in life when the backdrop of massive amounts of mental illness, ill health from the virus, isolation, grief, disruption to work, unemployment and so on are considered. I imagine the effects on university students to be particular and perhaps being forced to receive their education online only (as has been the case in many countries) has had important impacts, some of which are unlikely to be positive. it appears that thee conclusions made here do not take these other contextual factors into account
360 what do you mean improve individual values?
361 change 'pacifically' to 'specifically'
370 and on: the discussion moves from college students to 'young adults'. i think you need to define the age group you are looking at and it would be good to be consistent
4.4 as mentioned above, gender is another limitation of this study that it would be good to mention. it is also possible that future studies could build on your work and bring in more around mental health / mental illness, and maybe suggest some research that could be done to test ways that prosocial behaviour might be developed through the lens of awe.
overall i am not convinced that this study is connected well to the context of the pandemic - it is only loosely mentioned. it will be important to strengthen the links here and provide more thorough context of how this group experienced it, or remove the pandemic as a factor altogether. the study was not designed to test people before / during / after the pandemic (of course impossible unless coincidentally timed) so it is difficult to suggest that the pandemic has caused so many positive outcomes for college students.
Author Response
Thanks for your detailed suggestions. This help us to improve the manuscript and we have revised it.
Please see the attachment.

Reviewer 2 Report
This paper reports a study on the effects of awe on prosocial behavior and the mediating and moderating effects of other two psycho-social dimensions. Although the topic is interesting and the research is well designed in general, there are further clarifications, additions and reporting to be included in order to upscale the quality of the manuscript. Particularly, I am not convinced that the true added contribution of the research to the existing body of knowledge (as detailed below, the core model of relationships was already tested in a published study) is indeed as significant as that required for acceptance in a top journal as IJERPH.
- line 66 the phrase “it is fundamentally organized out of concern for others to enhance the welfare of others” is unclear, needs rewriting.
- line 82 “Awe-eliciting video likely induce a positive feeling and ultimately increase meaning in life” – this describes the result of a past research, and should be contextualized as such.
- other phrases that need rewriting: line 86 “more prosocial behavior people present”; line 116“Their studies have supported that” (the studies on what dimensions / concepts?)
- line 95 it’s unclear how “social activities can transcend the meaning in life”
- the contribution of the present study to the extant body of knowledge should be further clarified. Important in this respect, the Introduction mentions (line 98) “Lin reveals the mediating role of self-transcendent meaning in life between dispositional awe and prosocial behavior in his article, but the longitudinal study may be more convincing”. This implies that the central part of the model tested here (meaning in line as mediating the relation between awe and prosocial behavior) has already been investigated, and that the only addition of the present research is methodological, the investigation of these relations in a longitudinal fashion. This should be clarified, especially since the Discussion highlights a major contribution that the paper may have in this respect: “This study not only explains how awe affects prosocial behavior from a new theoret-357 ical perspective, but also expands the research field of prosocial behavior by accounting 358 for the above-discussed mediation mechanism”. Yet, since there are previously published studies on the exact same conceptual model, these assertion might be untrue.
- the paragraph from line 115, that aims to provide arguments supporting the idea that social support moderates the effect of awe on meaning in life, currently fails to reach this aim. It argues that awe could also negatively influence meaning in life (line 120), which indeed renders plausible the idea of a moderator of the effects of awe. But the rest of the paragraph does not provide a line of argumentation that would imply that social support can be such a moderator; instead, it offers evidence that social support is associated both to awe (line 125) and to social support (line 122). This is the typical argumentation of a mediation hypothesis, and not a moderation hypothesis. Thus, the latter is left unargued in the Introduction, and still requires evidence (theoretical and/or empirical) that would suggest that the effect of awe on meaning in life might differ in people with low social support in comparison to those with high social support. The first idea in the next paragraph (“perceived social support is a protective factor that reduces the sensitivity of meaning in life to negative predictors and increases the effect of positive predictors”) could be a good start for this attempt, but currently it’s just enounced and left unexplained.
- as the model tested includes both mediation and moderation effects, the results pertaining to both types of relationships should be extracted from the same PROCESS model (the complete one, which now is commented only with regard to moderation. Thus, the mediation results reported and commented should be those from this complete moderated mediation model as well.
- for both moderation analyses, the effects of the independent variable (i.e., awe, meaning in life) on the dependent should include (in the Figures and in the text) also the group with mean social support (located 1SD below and above the mean)
- in the first paragraph of section 4.3. the descriptions of the two moderating effects are reversed from those reported in the Results.
- line 372 the phrase “awe and perceived social support serve as substitutes” needs further clarification
- line 385 “If they have high awe working in conjunction with high levels of perceived social support, it would be more effective to maximize their prosocial behavior.” – this phrase implies the opposite to the results that is discussed here (the fact that the effect of awe is nonsignificant in people with high social support), which implies that increasing awe has no effect on prosocial behavior in this category of people.
- line 399 the explanation “that for college students with high perceived social support, 399 when they lack meaning, they can get comfort, encouragement and company from fami-400 lies and friends to compensate for the lack of presence of meaning in life caused by low awe.” should refer to the moderating role of social support in the relation between meaning in life and prosocial behavior. This part of the general model excludes awe, thus the explanations of the effects between these variables should not refer to awe.
- line 402 “a low level of perceived social support may make college students feel excluded from social relationships with others and reduce their caring about other things” – this idea could represent a potential explanation of the moderation effect (i.e., the weak effect of meaning in life in students with low social support), but it needs references to studies that would support the thesis that low social support “reduce caring about other things”.
- the next two phrases only describe the general moderation effect, thus they should not be placed at the end of this paragraph (where the actual explanation of the moderation should be), but instead in the beginning, merged with the current description.
- section 4.4. Limitations and future directions currently includes no future directions of investigation.
- another limitation is that the measurement of awe with a scale addressing dispositional traits towards experiencing emotions (and thus not affective reactions to current events or circumstances) renders the results less relevant to the COVID issue, as the manuscript repeatedly intends to persuade the reader. An instrument measuring the degree of awe in relation to the efforts of the medical staff in treating COVID patients, for instance, would have been appropriate to this aim.
Author Response
Thanks for your advices. We have revised it according to your suggestion.
Please see the attachment.
